# Predicting wave overtopping thresholds on coral reef-island shorelines with future sea-level rise

E. Beetham [1] & P.S. Kench [1,2]

Wave-driven flooding is a serious hazard on coral reef-fringed coastlines that will be exacerbated by global sea-level rise. Despite the global awareness of atoll island vulnerability, little is known about the physical processes that control wave induced flooding on reef environments. To resolve the primary controls on wave-driven flooding at present and future sea levels, we present a globally applicable method for calculating wave overtopping thresholds on reef coastlines. A unique dataset of 60,000 fully nonlinear wave transformation simulations representing a wide range of wave energy, morphology and sea levels conditions was analysed to develop a tool for exploring the future trajectory of atoll island vulnerability to sea-level rise. The proposed reef-island overtopping threshold (RIOT) provides a widely applicable first-order assessment of reef-coast vulnerability to wave hazards with sea-level. Future overtopping thresholds identified for different atoll islands reveal marked spatial variability and highlight distinct morphological characteristics that enhance coastal resilience.

[1] School of Environment, University of Auckland, Private Bag, 92010 Auckland, New Zealand. [2] Department of Earth Sciences, Simon Fraser University, 8888 University Drive Burnaby, British Columbia, Canada V5A 1S6. Correspondence and requests for materials should be addressed to E.B. (email: e.beetham@auckland.ac.nz)

Coral reefs effectively dissipate incident wave energy through a combination of breaking and friction[1] and fundamentally regulate how ocean waves interact with reef-fringed shorelines[2]. This 'natural breakwater' effect has enabled dense development on coastal margins and complete urbanisation on a number of atoll islands that are now highly vulnerable to impacts of future sea-level rise (SLR)[3,4]. Small and low-lying atoll islands are prone to flooding from spring tides[5], storm surge[6] and wave overtopping[7], with marine inundation predicted to become increasingly costly in the near future[3,4,8]. At present sea-level (SL), wave overtopping occurs every 2–3 years[8], with destructive flooding events every 10–30 years[7,9,10] allowing sufficient recovery time for agriculture, fresh-water resources and infrastructure. Global SL is predicted to rise by $0.44 \pm 0.17$ m (RCP2.6) to $0.74 \pm 0.23$ m (RPC8.5) by 2100[11], with a continued increase predicted for several centuries regardless of emission mitigation efforts[12,13]. SLR is a serious climate change hazard that has already impacted some reef-island systems[14] and is predicted to severely impact all reef-fringed and atoll island shorelines in the coming decades[15,16], affecting an estimated global population of 197 million people[4]. Unmitigated, SLR will increase the frequency and magnitude of wave-driven flooding events[8], eventually reaching a threshold when compounding events prevent recovery and compromise permanent habitation on coastal margins or entire islands[17]. Atoll-populations (>500,000 people across five nations) have limited land available for in-country relocation and may become climate change refugees when local adaptation efforts are exceeded[16,18]. However, future responses are unlikely to be uniform as there is considerable heterogeneity in island physical structure and energy exposure[19]. To assist local adaptation and timelines for future overtopping, there is a pressing need to resolve how the physical drivers of wave-driven flooding on reef coastlines vary spatially due to differences in wave climate exposure and island morphology. Here, we present a new model of wave overtopping that can be applied to predict SL and (or) wave height thresholds for overtopping on reef-fringed coastlines globally. The model is applied to a number of different atoll islands to understand how future inundation trajectories vary according to local morphology, wave climate exposure and potential geomorphic feedbacks.

Overtopping occurs when wave runup surges over the beach crest and across the island surface (Fig. 1a). Wave runup is a product of interactions between incident wave energy and reef morphology, with overtopping determined by the capacity of a shoreline to accommodate runup surges. Theoretically, runup levels and the occurrence of overtopping are, therefore, a predictable function of incident wave energy and first-order morphology characteristics that control dissipation. However, predicting runup and overtopping on reef coastlines is significantly complicated by a complex suite of nonlinear wave transformation processes that increase reef flat water level (setup) and produce energetic low-frequency (infragravity) wave surges[20]. Wave breaking on coral reefs creates a setup of mean water level across the reef flat[21] and the dissipation of incident wave groups drives a transfer of energy to long-period infragravity wave motions that surge across the reef flat[22]. Wave setup and infragravity waves are dynamically influenced by incident wave conditions and reef morphology and both significantly influence runup and overtopping on reef coastlines[20,23]. It is, therefore, essential that the nonlinear interactions between incident wave dissipation and these dynamic surf-zone processes are represented accurately when investigating runup and wave-driven flooding on reef coastlines[8,22].

Previous attempts to understand the physical drivers of future wave overtopping on reef coastlines focus on single case study

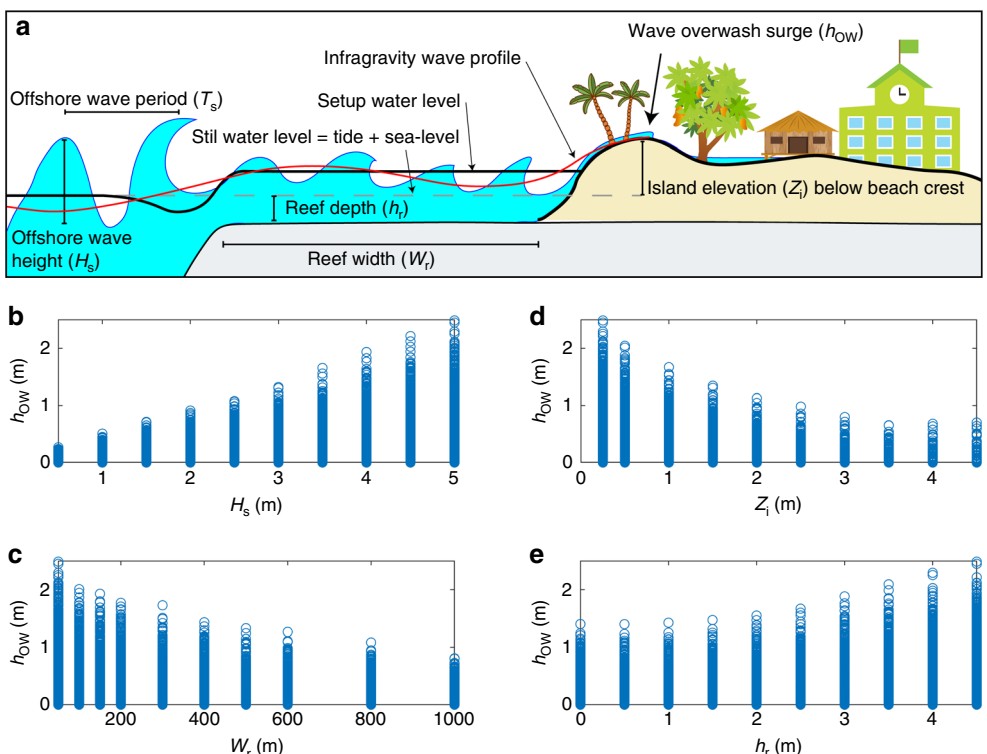

**Fig. 1** Physical controls on wave-driven flooding on atoll islands. **a** Conceptual diagram of primary variables that influence wave runup and overtopping on coral reef coastlines, highlighting the importance of wave setup and infragravity wave motions. **b–e** Model outputs showing the relationship between maximum wave overwash depth ($h_{ow}$) at the beach crest for the 19,020 simulations where overtopping was identified and individual forcing variables: Offshore significant wave height ($H_s$), island elevation above still water level ($Z_i$), reef width ($W_r$) and reef depth ($h_r$)

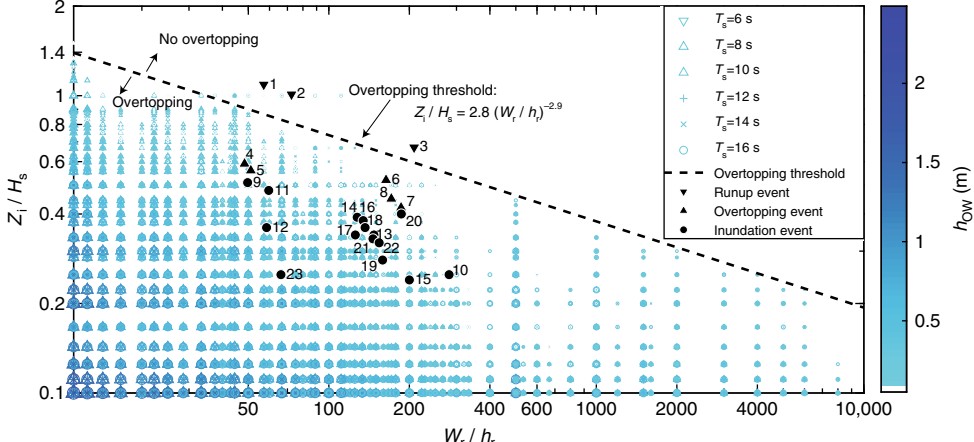

**Fig. 2** Threshold conditions for wave-driven flooding on atoll islands. Relationship between wave overwash depth, relative reef width ($W_r/h_r$) and relative island elevation ($Z_i/H_s$) for all modelled scenarios. Overwash magnitude is indicated by colour (colour bar) and marker size. The dashed black line delineates a threshold where no overtopping occurs (Eq. (1)). Note, there are 31 outliers where marginal overtopping ($h_{ow} < 0.02$ m) occurred above the threshold, out of the 3840 unique RRW ($n = 60$) and RIE ($n = 64$) combination. Black markers represent RRW and RIE combinations associated with documented events where significant inundation (circle), overtopping at the shoreline (squares) or high runup without overtopping (downward point triangle) occurred on a range of atoll islands (Table 1).

islands[8,17,24,25] that are not globally applicable, neglect overtopping magnitudes[26] or use simplified hydrodynamic models that do not resolve nonlinear surf-zone processes[17,27,28]. Our research advances previous work by applying a comprehensively evaluated physics-based numerical model[20,29] to directly simulate wave overtopping motions resulting from incident wave dissipation and nonlinear surf-zone processes. Our approach builds on previous studies that utilised a wide range of idealised bathymetries and wave conditions to investigate the impacts of SLR on reef coastlines[26–28,30]. Consequently, we present the first analysis of wave overtopping that is applicable to reef coastlines globally, under present and future SLs by representing 60,000 unique morphology, wave energy and SL scenarios.

## Results

**The reef-island overtopping threshold.** Of the 60,000 unique combinations of reef depth ($h_r$, below still water level), reef width ($W_r$), significant incident wave height ($H_s$), wave period ($T_s$) and island elevation ($Z_i$, above still water level) used in this analysis (Methods; Supplementary Table 1), wave overtopping occurred on 31.7% of the simulations, with maximum wave overwash depth ($h_{ow}$) at the beach crest ranging between 0.01 and 2.5 m (Fig. 1a). For simulations where overtopping was identified (Methods; Supplementary Fig. 1; Supplementary Data 1), a deterministic relationship was established between $h_{ow}$ and each forcing variable ($H_s$, $h_r$, $W_r$, $Z_i$), when all other factors were consistent (Fig. 1). Modelled $h_{ow}$ increased proportional to $H_s$ and $h_r$ (Fig. 1b, e), but decreased with increasing $Z_i$ and $W_r$ (Fig. 1b–e).

Subsequent analysis identified two new dimensionless parameters that collectively control the occurrence of wave overtopping on atoll islands (Fig. 2). Relative reef width (RRW) ($W_r/h_r$) quantifies the dissipative capacity of a reef and relative island elevation ($Z_i/H_s$) represents the potential for an island to withstand overtopping during a given event. RRW decreases with increasing SL and therefore represents how reefs will become less effective at dissipating incident wave energy, resulting in elevated runup levels at the shoreline. Low-relative island elevation (RIE) values (<0.6) indicate that an island is likely to be overtopped by a given event, unless RRW is high and the reef

is extremely dissipative. At present SL, overtopping occurs because high-wave events coincide with high tide, resulting in a low-RIE value. As SL increases, the associated decrease in $Z_i$ will lower RIE and therefore the incident wave energy required for overtopping will decrease. Across the spectrum of forcing and morphology conditions used in this analysis (Supplementary Table 1), RRW and RIE present a clear boundary that determines whether overtopping occurs on a reef-fringed shoreline (Fig. 2), which is defined by the reef-island overtopping threshold (RIOT; Eq. (1)), a simple function of island morphology and incident wave height:

$$\frac{Z_i}{H_s} = 2.8 \left(\frac{W_r}{h_r}\right)^{-0.29} \tag{1}$$

Modelled overwash magnitudes (Supplementary Data 2) were tested against RRW and RIE combinations from 23 published wave runup, overtopping and flooding events across five atolls and showed good agreement (Table 1). Elevated runup events (without overtopping) associated with either moderate waves at spring high tide (SHT), or large waves during a neap tide, are appropriately located above the overtopping threshold line (Fig. 2). Five events in the Marshall Islands that were associated with nuisance level overwash and minimal damage are all positioned below the overtopping threshold[7,23], with modelled inundation depths averaging $0.12 \pm 0.01$ m (Table 1). RRW and RIE combinations from fifteen significant island flooding events from the Marshall Islands[7,8,23], Taaku Atoll[31] and the Maldives[9] are located well below the inundation threshold, with modelled $h_{ow} = 0.24 \pm 0.02$ m, on average (Table 1). Of note, these significant island inundation events all occurred during high tide and were driven by either local storm activity or long-period swell with wave heights between 2 and 5 m (Table 1). Wave period is shown to have some influence on overwash magnitude (Fig. 1), but this is of a secondary importance compared to other input variables (Supplementary Fig. 2) and was, therefore, not included in the RIOT formula (Methods).

**Future overtopping trajectories with SLR.** The RIOT provides a powerful tool to explore and predict the vulnerability trajectory of

**Table 1 Documented wave runup, overtopping and inundation events used to compare model outputs**

| # | Island | Date | $H_s$ (m) | $W_r$ (m) | $h_r$ (m) | $Z_i$ (m) | Tide (m) | Event | Modelled Mean ± SE $h_{ow}$ (m) |
|---|---|---|---|---|---|---|---|---|---|
| *High runup* | | | | | | | | | |
| 1 | Fatato[20] | 23/06/2013 | 1.6 | 100 | 1.75 | 1.75 | 0.98 | RU, SHT | – |
| 2 | Fatato[20] | 23/06/2013 | 2.1 | 100 | 1.38 | 2.12 | 0.61 | RU | 0.03 ± 0.00 |
| 3 | Majuro-NE[7] | 28/11/2010 | 3 | 250 | 1.2 | 2 | 0.4 | W | – |
| *Overtopping* | | | | | | | | | |
| 4 | Majuro-S[7] | 8/10/2014 | 1.6 | 90 | 1.86 | 0.94 | 1.06 | N | 0.17 ± 0.01 |
| 5 | Majuro-S[7] | 29/06/2011 | 1.85 | 90 | 1.76 | 1.04 | 0.96 | N, SHT | 0.22 ± 0.02 |
| 6 | Roi-Namur NW[23] | 18/11/2012 | 2.2 | 270 | 1.65 | 1.15 | 0.9 | N, SHT | 0.07 ± 0.01 |
| 7 | Roi-Namur NW[23] | 9/12/2012 | 3.2 | 270 | 1.45 | 1.35 | 0.7 | N | 0.12 ± 0.02 |
| 8 | Majuro-NE[7] | 25/11/1982 | 3.84 | 250 | 1.46 | 1.74 | 0.66 | D, TC | 0.04 ± 0.01 |
| *Significant inundation* | | | | | | | | | |
| 9 | Majuro-S[7] | 24/06/2013 | 1.95 | 90 | 1.81 | 0.99 | 1.01 | D, S, SHT | 0.23 ± 0.01 |
| 10 | Takuu[31] | 10/12/2008 | 2.4 | 380 | 1.35 | 0.6 | 0.75 | D, E, S | 0.11 ± 0.01 |
| 11 | Majuro-S[7] | 8/06/1994 | 2.71 | 90 | 1.51 | 1.29 | 0.71 | D, S | 0.13 ± 0.01 |
| 12 | Majuro-S[7] | 17/11/1992 | 3.05 | 100 | 1.71 | 1.09 | 0.91 | TC, D | 0.38 ± 0.03 |
| 13 | Fares-Maathoda[9] | 12/05/2007 | 3.05 | 140 | 0.95 | 1.05 | 0.65 | D, E | 0.18 ± 0.01 |
| 14 | Majuro-NE[7] | 2/03/2014 | 3.17 | 250 | 1.96 | 1.24 | 1.16 | D, SHT | 0.22 ± 0.02 |
| 15 | Roi-Namur NE[8] | 8/12/2008 | 3.5 | 350 | 1.75 | 0.85 | 1 | D, E, SHT | 0.25 ± 0.01 |
| 16 | Majuro-NE[7] | 8/12/2008 | 3.5 | 250 | 1.86 | 1.34 | 1.06 | D, E, SHT | 0.20 ± 0.02 |
| 17 | Majuro-NE[7] | 17/11/1989 | 3.59 | 250 | 1.99 | 1.21 | 1.19 | D, SHT | 0.25 ± 0.01 |
| 18 | Majuro-NE[7] | 13/01/2011 | 3.76 | 250 | 1.83 | 1.37 | 1.03 | D, SHT, S | 0.26 ± 0.01 |
| 19 | Roi-Namur NW[23] | 2/03/2014 | 4 | 270 | 1.7 | 1.1 | 0.95 | D, E, SHT | 0.29 ± 0.03 |
| 20 | Majuro-NE[7] | 10/12/1997 | 4.11 | 250 | 1.34 | 1.66 | 0.54 | D, E, S, TC | 0.16 ± 0.01 |
| 21 | Majuro-NE[7] | 25/11/1979 | 4.5 | 250 | 1.71 | 1.49 | 0.91 | D, E | 0.18 ± 0.01 |
| 22 | Majuro-NE[7] | 8/01/1988 | 4.99 | 250 | 1.621 | 1.579 | 0.821 | TC, D | 0.28 ± 0.02 |
| 23 | Majuro-S[7] | 7/01/1992 | 5.18 | 100 | 1.51 | 1.29 | 0.71 | D, S, TC | 0.53 ± 0.02 |

RU: elevated runup without inundation; S: sediment deposited on island; N: nuisance level inundation with no significant damage; SHT: coincident with spring high tide; D: destruction to infrastructure; E: state of emergency, W: inundation warning issued; TC: tropical cyclone. Mean and standard error overwash depth ($h_{ow}$) are calculated from model outputs with the same RRW and RIE as each field scenario

different atoll islands to SLR, by adjusting $Z_i$ and $h_r$ magnitudes in accordance with how SLR will increase reef depth and decrease island elevation (Fig. 3). Trajectory curves are presented for a number of studied islands (from Table 1) using reef specific morphology characteristics and wave climate statistics for each islands 50 percentile (p50), 99 percentile (p99) and extreme significant wave height (Methods; Supplementary Table 2). Trajectory profiles highlight how SLR will exacerbate overtopping on atoll islands by compromising wave dissipation capacity (decreasing RRW), while also decreasing the magnitude of energy required for overtopping. It is important to note that the trajectories and thresholds presented here are based on simulations of nonlinear wave transformation on simplified reef geometries and do not represent the site-specific complexity of each individual reef-island system. This enables a first-order assessment of overtopping vulnerability across a wide spectrum of reef systems but does not represent site-specific variations in reef flat or beach-face topography and roughness. Therefore, the following assessment is appropriate for understanding trends and trajectories in reef-coast vulnerability and for identifying sites where higher resolution investigations are required.

Most islands do not experience overtopping at present SL when exposed to their respective p50 or p99 wave heights, although overtopping does occur on some lower elevation islands during p99 events at SHT (Fig. 3a, b). Significant overtopping is predicted to occur on the lower elevation islands of Nukutoa (Takuu), Fares-Maathoda and Majuro South under modal (p50) wave heights at SLR = 1 m. However, higher islands, including Fatato and Roi-Namur remain resilient to overtopping by mean waves at SLR = 1 m. All islands will experience overtopping or potential flooding under p99 wave

conditions with 1 m SLR, with major flooding predicted for each island during extreme wave events.

Overtopping trajectories indicate that an increase in reef depth associated with small steps in SL drives a substantial decrease in RRW and rapidly shifts island vulnerability towards the threshold for overtopping (Fig. 3). Once the threshold for overtopping is exceeded, the vulnerability trajectory for each island accelerates (drops rapidly) as RIE becomes the primary variable that determines the magnitude of potential flooding (Fig. 3). Significantly, our findings highlight that the capacity for islands to withstand SLR without wave-driven flooding is variable and strongly dependent on the antecedent morphology and wave climate of each island. In general, shallow reefs present a more rapid trajectory towards the overtopping threshold compared to deeper reefs, which is driven by a proportionally greater increase in wave energy at the shoreline. It is also evident that island elevation above SHT has a critical influence on future island vulnerability, however the timing and magnitude of future wave overtopping on islands of similar elevation will be controlled by differences in wave climate exposure and reef dissipation characteristics (Fig. 3).

Our results highlight that there is significant variation in the potential for atoll islands to withstand wave overtopping under future SLR scenarios, under the assumption that there is no adjustment to reef or island morphology. Predicted SLR thresholds for overtopping during modal wave conditions (p50 wave height specific to each reef) ranged between 0.1 m (Takuu) and 1.22 m (Fatato) above present SHT (Fig. 4a). This equates to a temporal range in the onset of regular overtopping occurring in the next few decades or beyond 2100. These results indicate that higher islands with a narrow reef (Fatato) are more resilient to flooding than low

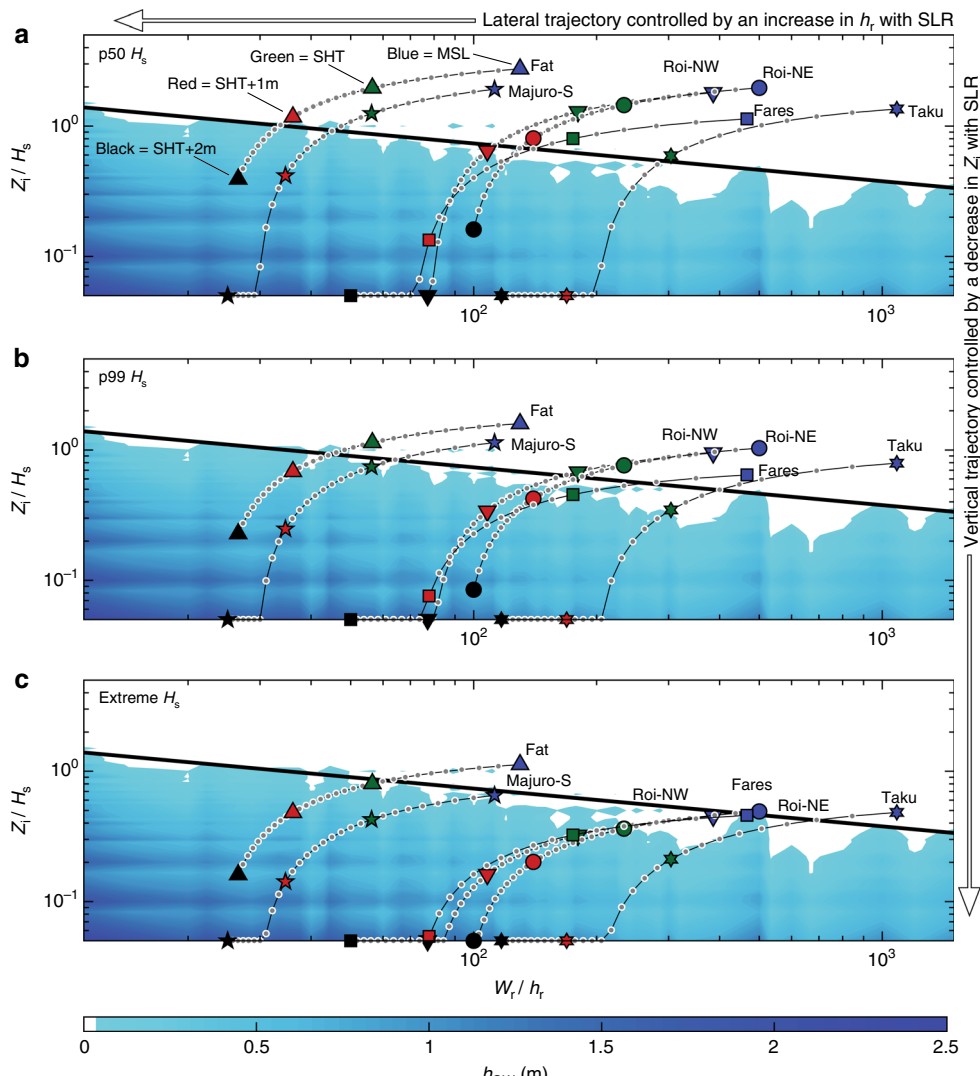

**Fig. 3** Inundation trajectories for different atoll islands and wave conditions. Inundation trajectories range from present mean sea-level (blue markers) to spring high tide (SHT; green marker), SLR = 1 m (red markers) and SLR = 2 m (black markers). **a** Fifty percentile wave height specific to each island (p50), **b** 99 percentile wave height (p90) and **c** extreme incident waves (average of the top 10 wave heights over 30 years). Trajectory curves account for how SLR will modify both axis because island elevation decreases ($y$-axis) when reef depth increase ($x$-axis), assuming no morphological adaptation. Overwash magnitudes (colour bar) for each combination of RRW and RIE, with associated standard error are the same in each plot and are presented in Supplementary Data 2

islands with a wide reef (Taaku). Differences between extreme and mean wave heights within a local wave climate also exert an important control on inundation vulnerability. For example, although Roi-Namur exhibits a high SLR threshold for inundation under modal wave conditions, it is one of the most vulnerable islands during energetic events because of its exposure to comparatively large 99.9 percentile (3.7 m) and extreme (6.2 m) waves (Supplementary Data 2; Methods). A key impact of SLR in the context of flood exposure on reef coastlines is that less energy will be required for overtopping, but more energy will be available at the shoreline. All the islands examined (Table 1) require $H_s \geq$ 1.4 m for overtopping at contemporary SHT, with $H_s > 2.8$ m required for overtopping on Fatato and Roi-Namur (Fig. 4b). The predicted threshold wave height for overtopping decreases by 69% (on average) for SLR = 1 m at SHT, with the greatest reduction noted for Takuu (100%; SL inundated), Fares (86%) and Majuro (71%), compared to Roi-Namur (54%) and Fatato (47%).

**Overtopping thresholds and geomorphic feedbacks.** Numerical modelling simulations and the subsequent analysis of future

overtopping trajectories were undertaken based on the conservative assumptions that island morphology and reef structure remain static over decadal timescales. Such assumptions suggests that model outputs yield worst-case overtopping scenarios by omitting two important morphological feedbacks that have significant potential to mitigate the flood susceptibility of islands. First, vertical accretion of the outer reef flat in response to new accommodation space for coral growth, which in some areas is predicted to exceed rates of SLR associated with RPC6.0 (7.4 ± 2.8 mm y⁻¹)[32]. Using Eq. (1) to quantify SLR thresholds for overtopping with theoretical keep-up ($\Delta h_r = 0$) and catch-up reef ($\Delta h_r = 0.5\Delta SL$) responses (Methods) identifies that future reef growth will delay the onset of flooding by raising SLR thresholds for overtopping by an average of 0.11 ± 0.01 m for keep-up reefs and 0.08 ± 0.01 m for catch-up reef responses (Fig. 4c). Significantly, future reef accretion has some potential to delay the impacts of SLR on atoll islands but has limited capacity to prevent overtopping, especially during higher energy events (Fig. 4). Reef accretion is effective because it limits the decrease in RRW but it cannot protect islands from overtopping because RIE still

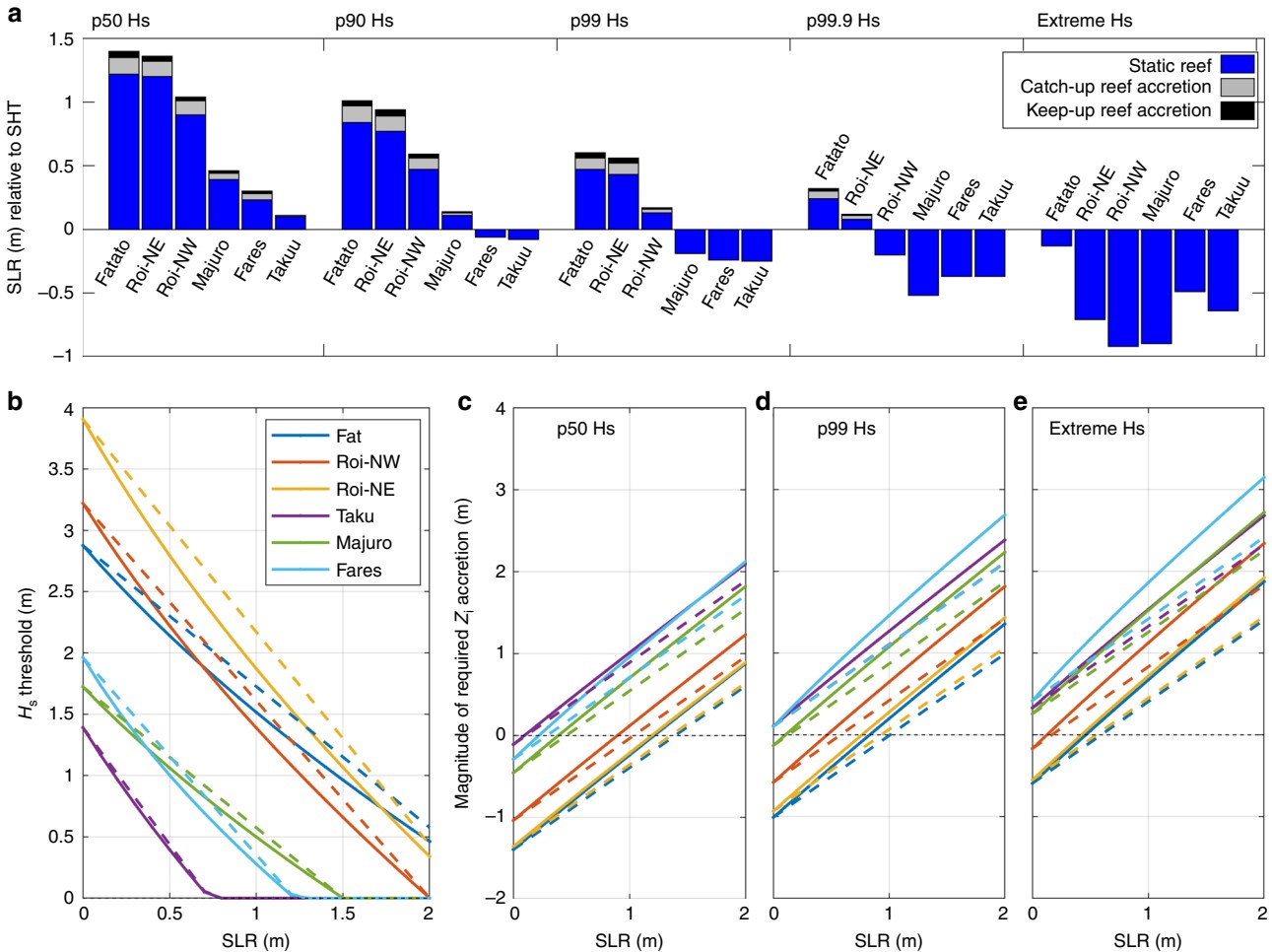

**Fig. 4** Threshold conditions for wave overtopping on atoll islands with different reef response scenarios. **a** Sea-level rise thresholds for wave overtopping on different atoll islands with no reef response, a catch-up and a keep-up reef responses, for different wave conditions that are specific to each island. **b** Threshold incident wave heights that will cause overtopping at spring high tide on different atoll islands for a range of future SLR magnitudes. Solid lines represent no reef adjustment and dashed lines represent a keep-up reef response. **c–e** The magnitude of vertical island accretion required to prevent wave overtopping at different magnitudes of SLR for different wave conditions. Note that the average gradient of solid lines (no reef response) is 1.2 compared to a gradient of 1.0 for dashed lines (keep-up reef response)

decreases with SLR. Therefore, wave overtopping will still occur on islands with a keep-up reef, but the magnitude of potential flooding will be more manageable.

Second, island morphodynamic feedbacks will likely occur as SLR allows wave overwash processes to transport sediment from the beach-face to the island surface, resulting in an increase in island elevation and lagoonward migration[33]. We do not consider how potential adjustments to island shoreline morphology will change future overtopping thresholds, but we do apply the RIOT formula to calculate how much vertical accretion is required to prevent overtopping, assuming no change in reef width. Eq. (1) was used to predict the magnitude of vertical island adjustment required to prevent overtopping at different SLs, wave events and reef responses (Fig. 4c–e). Once the SLR threshold for over-topping is reached, vertical accretion of the island shoreline will need to exceed the rate of SLR by an average of at least 20% ($\Delta Z_i = 1.2\Delta SL$) to maintain a shoreline elevation at equilibrium with the overtopping threshold associated with a give-up reef response (Fig. 4c–e). This proportionally higher increase in $Z_i$ is required to compensate for the decrease in $h_r$ that allows higher waves to reach the shoreline. Accordingly, island accretion requirements with a keep-up reef response are equal to the

magnitude of SLR. Since many of the islands in Table 1 currently have some capacity to prevent overwash under moderate waves at low magnitudes of SLR, the average vertical island accretion required to prevent overtopping with 1 m of SLR and p99 waves is 0.82 m, notably less than the magnitude of SLR. However, the magnitude of adaptation varies widely, with Takuu and Fares-Maathoda both needing 1.3–1.5 m of vertical island accretion to prevent overtopping at SLR = 1 m (Fig. 4c).

**Implications for atoll island vulnerability.** Our results present valuable new insight on the physical drivers of wave inundation on coral reef coastlines and provide a widely applicable tool for predicting thresholds for wave overtopping driven by extreme events or future SLR. Our results indicate that the lowest islands examined will suffer from chronic wave overtopping with small increases in SL (0.1–0.4 m) that are predicted to occur before 2100[11], assuming no eco-morphodynamic adaptation to reef or island structure. We also identify characteristics that promote resilience to inundation with SLR, allowing some islands to potentially remain habitable beyond 2100. Consequently, our research reveals that the impacts of SL rise on atoll islands will be highly variable across different reef systems and highlights the

importance of antecedent morphology in determining future vulnerability trajectories. We resolve the nonlinear balance between boundary forcing and dissipation variables that collectively determine the occurrence of wave overtopping on reef-fringed coastlines by deriving an idealised formula to quantify thresholds for wave-driven flooding that is applicable to a wide range of reef-island settings. The RIOT can be applied in coastal and climate change management practise to identify inundation-prone areas and locations with high natural resilience, based on easily measured morphological characteristics. Such knowledge enables a strategic and site-specific foundation to plan for and mitigate the impacts of future flooding events.

## Methods

**Numerical model**. A previously evaluated fully nonlinear numerical model was used to simulate wave shoaling, transformation, runup, overtopping and inundation for this research. The open-source Green–Naghdi solver[34] provides a fully nonlinear, but weakly dispersive solution for the Boussinesq equations that is applicable to large amplitude waves interacting with complex morphologies, such as coral reefs. A shock-capturing nonlinear shallow water solution is applied in areas of wave breaking and runup to ensure areas of rapid dissipation are handled with stability and accuracy[34,35]. The phase-resolving model we apply here has been comprehensively evaluated for accurately simulating wave dissipation, infragravity wave behaviour, wave setup, runup and overtopping on coral reef environments, using field data[20] and wave flume[29] experiments. The documented open-source model is available online[36], with existing literature providing more information on the numerical scheme[34] and its application to simulate waves on coral reefs[20,25].

**Numerical simulations**. A total of 60,000 unique simulations were undertaken for this analysis (Supplementary Table 1). These combinations comprise 60 boundary conditions (10 $H_s$ variations and 6 $T_s$ variations) and 1000 morphology variations (10 $W_r$ variations, 10 $h_r$ variations and 10 $Z_i$ variations) that collectively represent a global spectrum of atoll islands and energy settings. Consistent values were used for beach slope (0.1) and reef slope (0.14) because these variables are known to have a deterministic but secondary influence on wave runup elevation[26,27]. Friction is known to vary on coral reefs, depending on ecological and sedimentary characteristics. We account for friction using an implicit quadratic bottom friction equation with a coefficient of $C_f = 0.04$, representative of a moderately rough reef environment[20]. We do not account for variability in roughness by adjusting $C_f$ because this has a deterministic influence on runup and is of secondary importance compared to changes in water depth[26]. The contribution of possible future reductions in friction associated with declining structural reef complexity to overtopping scenarios is a subject recommended for future study. To represent wave breaking in the numerical model, a threshold free-surface slope was used to split between dispersive and dissipative equations, with a slope threshold of 1 based on previous work[20]. Modelled wave conditions were generated with a JONSWAP spectrum, using $\gamma = 3.3$ and each simulation was run for 35 min of wave activity to ensure that a saturated wave field developed on the reef platform, allowing a number of wave groups to potentially overwash the island. Waves released at the offshore boundary propagated across a 60 m deep shelf for 200 m before interacting with the moderately steep reef slope, reef flat and island (Supplementary Fig. 1). To maintain consistency with previous model calibration simulations[20], a uniform $\Delta x$ of 1 m was used to discretise the model domain. The island profile was designed to ensure that maximum elevation (input $Z_i$ value) remained constant for 50 m before a leeward beach slope decreased island elevation to zero (Supplementary Fig. 1). This leeward section was sufficient for overwashed water to pool without reflecting back onto the ocean reef flat. All simulations for this research were computed on the NeSI Pan cluster, through the University of Auckland.

**Data analysis**. Wave overwash was identified by extracting time-series data at 10 Hz at a location on the island berm positioned 70 m landward of the beach toe (Supplementary Fig. 1). Time-series data after 512 s was used to calculate mean and maximum overwash depth for each simulation. Overwash outputs for each simulation are presented in Supplementary Data 1, alongside input conditions. To establish a representative value of overwash depth for each RRW and RIE combination, the top third of all values for maximum overwash depth that had the same RRW and RIE combination were averaged (Supplementary Data 2). The same data points were used to calculate standard error, which are also presented in Supplementary Data 2, along with the number of points used to calculate averages (*n*). When calculating RRW, a minimum reef depth of 0.1 was used to avoid dividing by zero for dry reef scenarios.

The influence of wave period on overtopping is not accounted for in the RIOT formula. Wave period is known to exert some influence on runup magnitude and the relationship between $T_s$ and overwash depth was investigated for this analysis.

Based on our numerical outputs, very short period waves (6 s) result in reduced overwash magnitudes compared to moderate (8–12 s) and long (14–16 s) period waves. Overall, the influence of wave period on overwash occurrence and magnitude was of a secondary order compared to wave height and island elevation, which became the focus of subsequent analysis (Supplementary Fig. 2).

**Wave climate statistics**. Hourly wave climate data for 30 years (1980–2009), from the CSIRO Wave Watch 3 hindcast database[37], were used to calculate wave climate statistics for Fatato (179°E; −8.5°N), Roi-Namur (167°E; 9.4°N), Majuro (171°E; 6.9°N), Takuu (158°E; −4.4°N), and Fares-Maathoda (72.8°E; 0°N). Time-series wave data were filtered to only include wave events where peak wave direction was travelling within 90 degrees of shore-normal to the island. Directionally filtered wave heights were used to calculate 50, 99, 99.9 and 99.99 percentile wave heights, and extreme wave heights, calculated as the average of the top 10 wave heights over the 30 years. Wave height statistics for each example island were used to calculate inundation trajectories and future SL thresholds for wave inundation (Supplementary Table 2).

**SL thresholds**. SLR thresholds for wave inundation were calculated for each island and wave statistic by incrementally adjusting $Z_i$ and $h_r$ at small (0.001 m) steps that represent increases in SL. Thresholds were identified by iterating the increase in SL until the calculated $H_s$ threshold for overtopping (using Eq. (1)) was less than the wave height statistic being examined. SLR thresholds were calculated for three reef response morphologies, allowing us to quantify how effective future reef accretion may be at mitigating SLR impacts. To represent no change in reef structure (give-up response), the increase in reef depth ($\Delta h_r$) was equal to the increase in SL ($\Delta SL$). To represent a catch-up reef response, where the rate of reef accretion is less than the rate of SLR, the increase in reef depth was equal to half the increase in SL at each increment. To represent a keep-up reef response, where carbonate accumulation processes allow a reef to accrete vertically at the same rate as SLR, the change in reef depth was zero. Given the significant uncertainty and regional variability regarding the rate and magnitude of future SLR[12], we chose not to convert our identified SLR thresholds into temporal dates.

**Limitations**. Simulations used to develop the RIOT were undertaken using a fully nonlinear Boussinesq-type model that has been comprehensively evaluated for simulating wave processes on coral reefs. However, there are limitations associated with numerical simulations and data synthesis that need to be understood before Eq. (1) can be applied in practise. First, using a one-dimensional across-reef profile omits the influence of wave refraction that is important on some reefs. Second, using an idealised island and reef morphology in each simulation omits the influence of micro-topography on wave processes on the reef flat and beach-face. Accordingly, our findings are based on a simplified geometry that represents the broad morphological characteristics of a reef. Third, the hydrodynamic model does not account for porosity, sediment transport or morphodynamic change which means our inundation values likely present a worst-case scenario. Fourth, we do not represent variations in friction, reef slope and beach slope that are known to influence runup on coral reef shorelines because we consider these to be second-order variables. Fifth, the location of the threshold curve that defines whether inundation occurs or not for a given combination of RIE and RRW was defined visually and accounts for 99.2% of all combinations associated with overtopping or inundation.

Despite these limitations, the placement of RRW and RIE combinations for 23 known inundation, overtopping and runup events (Fig. 2), provides confidence that these limitations are acceptable on many atoll settings. The motivation for this analysis was to develop a first-order method for rapid assessment of island vulnerability to overtopping, which requires focusing on key variables and omitting local factors such as reef slope and friction that have a secondary influence. Therefore, the RIOT is most applicable for wide-scale and general assessments of island vulnerability and can identify areas where more specific investigations can be made, if required.

**Code availability**. The Green–Naghdi solver used in this research is part of the open-source Basilisk software package. The documented source code is available online (http://basilisk.fr/src/green-naghdi.h) and the software is freely available to install and use. Linux shell scripts that were used to pre-process model runs and Matlab codes that were used to process model outputs are available from the corresponding author (e.beetham@auckland.ac.nz) on request.

## Data availability

All data used in the research are included as Supplementary Information or Supplementary Data. Any other information related to this article can be requested from the corresponding author (E.B.): e.beetham@auckland.ac.nz.

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

## Acknowledgements

We thank the New Zealand eScience Infrastructure (NeSI) High Performance Computing facilities for providing us with the necessary 26,400 simulation hours required for this research. We specifically thank Sina Masoud-Ansari, at the Centre for eResearch, University of Auckland, for providing excellent technical support. NeSI is funded jointly by the University of Auckland, other collaborator institutions and the New Zealand Ministry of Business, Innovation and Employment. Special thanks to Tracey Turner for helpful discussions and manuscript edits. Tree and building art in Fig. 1 were obtained from vecteezy.com under standard licence for reuse.

## Author contributions

E.B. conceived the study, undertook the numerical modelling simulations, analysed the data and prepared figures. E.B. and P.K. interpreted the data and wrote the manuscript.

## Additional information

**Competing interests:** The authors declare no competing interests.

