## [Peer Review File · Nature Communications]

Reviewer #2 (Remarks to the Author):

This is a well written paper on a topical subject. The title is perhaps a bit ambitious given that it is an idealised study and overwash depths do not really represent flooding. There are a number of issues that need discussing more fully, but these do not require any further modelling. However, having said that, this paper is a considerable advance on another recent paper in NC that claimed to indicate island flood levels on the basis of R2% runup elevations (which has no basis in physics whatsoever). So, I support publication subject to the points below being addressed by the authors.

Lines 18-21 imply use as a rapid assessment tool. As an engineer I think this needs some caveat. Real locations have complexity that is not included here, and thus relying on such a tool, or implying it is suitable for such an analysis in a local context, in my view risks overstating the protection afforded by reefs, particularly in extremes. Thus, I think emphasising the trajectories and general science outcomes is more beneficial.

The approach taken is well described and appropriate for the idealised scenario. But, it should be mentioned that the geometry considered is highly simplified as the authors know well, both for the reef and the "beach", where coral ridges and rubble are often built and maintained by waves as illustrated in the authors' own publications, and there are no mangroves etc. There is no point glossing over this and it is not really provided in lines 422-430. What do the reef-beaches of the specific islands considered really look like? This does not negate the benefit of the study, but limits use to understanding broad scale trends, not site specific conditions, or only a very restricted set of sites.

In lines 70-78, it should be noted that the approach taken, of simulating a large range of idealised bathymetry and wave conditions using a physics-based wave model, was first proposed to study the protection afforded by coral reefs in this same context by Baldock et al. 2014; 2015 (Marine Pollution Bulletin), and included an estimate of how the morphological characteristics of the reef system might impact on sediment transport and beach erosion.

The issue of how beach or beach crest morphology might change over time, and how this impacts the study conclusions, is briefly considered, lines 223-225, but again I believe the authors' own work on reef island morphology indicates that greater discussion of caveats is needed. These, or at least many other, islands have wave-built beach crests. There is little reason to suppose that they cannot keep up with rising sea levels (even if that means transgression). For example, during overwash, beach berms can easily keep up with the rise from neap to spring tides on natural beaches (Baldock et al, 2008, Geomorphology), so the response time is very rapid compared to sea level rise. Thus, it needs to be very clearly emphasised that the flood trajectories are for no change in current beach

crest elevations. This issue has been discussed for reef crest elevation, but not sufficiently for berm crest elevation.

The island inundation index is an interesting new function, but is inundation not a bit overstated? There is only a threshold, and the index does not indicate any magnitudes. Overtopping index?

Line 236. I think critical new insight is rather over doing it. Google can tell me the Marshall Islands are suffering from rising seas (or reef degradation?) and have been for some time. Islands that are 1-2m above sea level and currently often inundated at highest tides will clearly be impacted by 0.4m sea level rise. And to say that elevation is the single most critical value determining the magnitude of overtopping is rather obvious to a layperson, let alone coastal scientists.

The reef geometry is an important control on what will happen, and the changes in nearshore conditions are a complex function of this geometry. This has been well illustrated in the previous work of Baldock et al. noted above, and they are not often monotonic functions. I think the authors could also illustrate how wave period changes the picture, since I would expect this to be very important, and it is very difficult to judge from figure 2 what that might be. Indeed, since neither parameter in figure 2 includes wave period, why is it there? A plot of h versus T for a given W/h would be more instructive.

Tom Baldock, July 2018

Reviewer #3 (Remarks to the Author):

The manuscript presents a tool for calculating wave overtopping thresholds on reef coastlines based on a combination of fully nonlinear wave transformation numerical simulations representing a wide range of wave energy, morphology, and sea-level conditions. The authors identify that little is known about the actual physical drivers of wave inundation on reef coasts or how sea-level rise will modify thresholds for inundation. They claim that their findings provide new insights on the physical drivers and a tool for a first-order assessment of the impacts of sea-level rise and future vulnerability trajectories of atoll islands globally.

The proposed island inundation index is novel and a great contribution that builds on the non-dimensional reef energy window index initially proposed by Kench et al. (2006). The tool will be of interest to both experts and the wider community.

The physics-based hydrodynamic numerical model used in this study resolves nonlinear surf-zone processes on coral reefs, as opposed to other recent research based on simpler models (e.g. Storlazzi et al. (2018). Most atolls will be uninhabitable by the mid-21st century because of sea-level rise exacerbating wave-driven flooding. *Science Advances*). The assessment of the model is outside my scope of expertise but has been previously evaluated using field data and wave flume experiments. The applicability and robustness of the model is acceptable for this application as evidenced by the testing against documented events. Furthermore, both the model and data are available online, making the approach reproducible.

Overall, I think that the proposed work is quite valuable as a first-pass assessment of island vulnerability to inundation. In addition to helping to identify hot-spots and islands at risk, this manuscript also identifies two important morphological feedbacks (reef accretion and island dynamics), secondary variables (friction variability, reef slope) and incorporating refraction (expanding the model to 2D) as caveats/limitations that need further exploration and can help directing future research agendas.

I recommend publications after addressing the following minor revisions:

L8 Wave-driven flooding

L27 Consider rewording this sentence as too general (e.g. mangroves also regulate interactions between ocean waves and tropical coastlines).

L38 Reword to include that SLR has already impacted low-lying coral reef islands (e.g. Albert et al. 2016. Interactions between sea-level rise and wave exposure on reef island dynamics in the Solomon Islands. *Environmental Research Letters*, 11(5), 054011).

L48 Suggest including: Barnard et al. (2015). Coastal vulnerability across the Pacific dominated by El Nino/Southern Oscillation. *Nature Geosci*, 8, 801-807. doi:10.1038/ngeo2539

L194 What about scenarios of island or reef erosion?

L372 Recent work by Harris et al. (2018). Coral reef structural complexity provides important coastal protection from waves under rising sea levels. *Science Advances*, 4(2). doi:10.1126/sciadv.aao4350 concludes that structural complexity on reef flats is more important than sea-level rise in terms of dissipation when it comes to average wave heights based on field data. Suggest rewording this

sentence to reflect that not accounting for variability in roughness is a caveat of the model and needs to be incorporated in future studies.

Javier Leon

School of Science and Engineering, University of the Sunshine Coast

Response to reviewer #2

“This is a well written paper on a topical subject. The title is perhaps a bit ambitious given that it is an idealised study and overwash depths do not really represent flooding. There are a number of issues that need discussing more fully, but these do not require any further modelling. However, having said that, this paper is a considerable advance on another recent paper in NC that claimed to indicate island flood levels on the basis of R2% runup elevations (which has no basis in physics whatsoever). So, I support publication subject to the points below being addressed by the authors.”

We thank the reviewer for providing complementary feedback on our research and for supporting publication following revisions. We agree that the submitted title is not entirely suitable, and have changed this to better represent the substance of the paper. The new title is:

“Predicting wave overtopping thresholds on coral reef island shorelines with future sea-level rise”

“Lines 18-21 imply use as a rapid assessment tool. As an engineer I think this needs some caveat. Real locations have complexity that is not included here, and thus relying on such a tool, or implying it is suitable for such an analysis in a local context, in my view risks overstating the protection afforded by reefs, particularly in extremes. Thus, I think emphasising the trajectories and general science outcomes is more beneficial.”

This is a fair point and we understand that our proposed method is limited in that it does not account for site specific nuances that have complexity beyond what we represent. To modify this sentence, we have changed the sentence to ‘provide a first-order’ assessment of vulnerability. This wording implies that we don’t account for all localised factors.

“The approach taken is well described and appropriate for the idealised scenario. But, it should be mentioned that the geometry considered is highly simplified as the authors know well, both for the reef and the "beach", where coral ridges and rubble are often built and maintained by waves as illustrated in the authors’ own publications, and there are no mangroves etc. There is no point glossing over this and it is not really provided in lines 422-430. What do the reef-beaches of the specific islands considered really look like? This does not negate the benefit of the study, but limits use to understanding broad scale trends, not site specific conditions, or only a very restricted set of sites.”

This is a good point. We agree that this simplification is not explicitly outlined. In addition to the changes we made to lines 125-132 in revised text (outlined in response to the editor comments) we have added the following new point to the list of limitations that addresses this comment:

“Using idealised island and reef morphologies omits the influence of micro-topography on wave processes on the reef flat and beach face. Accordingly, our

findings are based on a simplified geometry that represents the broad morphological characteristics of a reef.”

“In lines 70-78, it should be noted that the approach taken, of simulating a large range of idealised bathymetry and wave conditions using a physics-based wave model, was first proposed to study the protection afforded by coral reefs in this same context by Baldock et al. 2014; 2015 (Marine Pollution Bulletin), and included an estimate of how the morphological characteristics of the reef system might impact on sediment transport and beach erosion.”

We have added the two recommended papers, along with a few others that utilise a batch-processing approach to obtain synthesised knowledge from large sets of numerical modelling experiments. The following sentence has been added (revised text lines 73-75), with appropriate references: “Our approach builds on previous studies that utilised a wide range of idealised bathymetries and wave conditions to investigate the impacts of sea-level rise on reef coastlines”.

“The issue of how beach or beach crest morphology might change over time, and how this impacts the study conclusions, is briefly considered, lines 223-225, but again I believe the authors' own work on reef island morphology indicates that greater discussion of caveats is needed. These, or at least many other, islands have wave-built beach crests. There is little reason to suppose that they cannot keep up with rising sea levels (even if that means transgression). For example, during overwash, beach berms can easily keep up with the rise from neap to spring tides on natural beaches (Baldock et al, 20008, Geomorphology), so the response time is very rapid compared to sea level rise. Thus, it needs to be very clearly emphasised that the flood trajectories are for no change in current beach crest elevations. This issue has been discussed for reef crest elevation, but not sufficiently for berm crest elevation.”

As the reviewer points out, the submitted manuscript included a brief comment on the morphodynamic adaptation of reef shorelines to future sea level. We wanted to acknowledge this point but not discuss it in detail because understanding morphodynamic feedbacks is not the objective of this paper, but it is the specific focus of other work currently in preparation. To acknowledge this point we have made minor edits throughout the manuscript to ensure the

reader is aware that all overtopping trajectories and thresholds are clearly for an island system that undergoes no morphological change.

“The island inundation index is an interesting new function, but is inundation not a bit overstated? There is only a threshold, and the index does not indicate any magnitudes. Overtopping index?”

On reflection, we agree. We have changed the island inundation index to the reef island overtopping threshold (RIOT).

“Line 236. I think critical new insight is rather over doing it. Google can tell me the Marshall Islands are suffering from rising seas (or reef degradation?) and have been for some time. Islands that are 1-2m above sea level and currently often inundated at highest tides will clearly be impacted by 0.4m sea level rise. And to say that elevation is the single most critical value determining the magnitude of overtopping is rather obvious to a layperson, let alone coastal scientists.”

Contrary to what the reviewer suggests, we maintain that our manuscript generates critical new insight on the future vulnerability of reef coastlines to sea level rise. We have clarified the wording of the final paragraph to emphasise what we believe are critical new insights that result from our research. Importantly, this manuscript is the first to assess how interactions between primary boundary conditions (wave climate, sea level) and reef structural characteristics (depth, width and island height) influence overtopping on reef coastlines. While each of these variables is known to have a deterministic influence on wave overtopping, our research shows that it is possible to account for variability in all of these parameters.

To suggest that island vulnerability is controlled by elevation alone is misleading and significantly oversimplifies our outputs. One of our key outcomes is that future island vulnerability will vary spatially according to differences in morphology, which counters the assumption from previous work that case study insight is transferable. For example, our work shows that two islands of the same elevation may have significantly different vulnerabilities to future SLR because of differences in either reef depth, width or wave climate. Such

differences become significant in planning for an increase in wave driven flooding and elevated risk to communities.

“The reef geometry is an important control on what will happen, and the changes in nearshore conditions are a complex function of this geometry. This has been well illustrated in the previous work of Baldock et al. noted above, and they are not often monotonic functions. I think the authors could also illustrate how wave period changes the picture, since I would expect this to be very important, and it is very difficult to judge from figure 2 what that might be. Indeed, since neither parameter in figure 2 includes wave period, why is it there? A plot of how versus T for a given W/h would be more instructive.”

This is a good point and we acknowledge that the influence of wave period was not discussed in the manuscript but agree that it should be. The main reason for omitting wave period in Equation 1 was that our results show that wave period had a secondary influence on overtopping compared to other parameters. To amend this, we have included a new Supplementary figure (Sup Fig 2) as suggested, and added a new comment in the Methods section that outlines the influence of wave period when applying Equation 1. Supplementary Figure 2 shows that wave periods of 6 s are associated with significantly lower overwash levels compared to all other periods (as expected). There is some difference between moderate periods waves (8-12 s) and long period swell (14-16s) but as shown as a comparison in Supplementary Figure 2, this relationship between wave period and overwash depth is significantly weaker than the relationship between overwash depth and H_s or Z_r . Therefore, we chose to focus the analysis using only primary variables that are appropriate for most or all open ocean wave periods incident to reef islands.

Reviewer #3

"I recommend publications after addressing the following minor revisions:

Thank you for providing complementary feedback and a suggestions for improving our paper.

L8 Wave-driven flooding"

Typo corrected.

"L27 Consider rewording this sentence as too general (e.g. mangroves also regulate interactions between ocean waves and tropical coastlines)."

Sentence was changed to specifically address coral reef fringed shorelines.

"L38 Reword to include that SLR has already impacted low-lying coral reef islands (e.g. Albert et al. 2016. Interactions between sea-level rise and wave exposure on reef island dynamics in the Solomon Islands. Environmental Research Letters, 11(5), 054011)."

Sentence was modified to include the recommended reference:

"SLR is a serious climate change hazard that has already impacted some reef island systems¹⁴ and is predicted to impact all reef-fringed and atoll island shorelines in the coming decades^{15,16}, affecting an estimated global population of 197 million people⁴."

"L48 Suggest including: Barnard et al. (2015). Coastal vulnerability across the Pacific dominated by El Nino/Southern Oscillation. Nature Geosci, 8, 801-807. doi:10.1038/ngeo2539."

After carefully considering the recommended reference we decided it was not directly relevant for the suggested sentence or wider manuscript because it is not specific to reef environments.

"L194 What about scenarios of island or reef erosion?"

No change has been made.

Future scenarios of reef and island erosion are controversial but important topics that we believe are beyond the scope of our manuscript. Our focus was to present overtopping thresholds for the simplest scenario (no change) and to discuss how potential feedbacks that are relevant for management applications (accretion of the reef or island berm) may offset or delay future risk. In regards to reef accretion, our analysis was advised by well understood scenarios of reef responses to sea level rise (keep-up and catch-up modes). No such theory is available for representing scenarios of reef erosion and we believe that any future work on this topic needs to be driven by field observations and focused effort. Regarding changes in island elevation, most existing discussions on the morphological adaptation of islands and barriers to sea level rise show there is strong potential for accretion, with a lower probability of erosion (Baldock et al, 20008, Geomorphology). We chose to focus on the magnitude of island berm accretion required to prevent overtopping because this variable is a proxy for runup elevation and is relevant for guiding management options. We believe it is not possible to accurately represent scenarios of island erosion without undertaking additional research to quantify erosion rates which are not yet readily available. To address some of these existing research gaps, future work will focus on exploring reef island flooding in the context of sea-level rise and morphodynamic change.

“L372 Recent work by Harris et al. (2018). Coral reef structural complexity provides important coastal protection from waves under rising sea levels. Science Advances, 4(2). doi:10.1126/sciadv.aao4350) concludes that structural complexity on reef flats is more important than sea-level rise in terms of dissipation when it comes to average wave heights based on field data. Suggest rewording this sentence to reflect that not accounting for variability in roughness is a caveat of the model and needs to be incorporated in future studies.”

As suggested by the reviewer we have added a comment in the methods to state that our omission of not representing variable friction magnitudes is a simplification that should be addressed in future studies: “The contribution of possible future reductions in friction associated with declining structural reef complexity to overtopping scenarios is a subject recommended for future study.”

Response to reviewer #2

“This is a well written paper on a topical subject. The title is perhaps a bit ambitious given that it is an idealised study and overwash depths do not really represent flooding. There are a number of issues that need discussing more fully, but these do not require any further modelling. However, having said that, this paper is a considerable advance on another recent paper in NC that claimed to indicate island flood levels on the basis of R2% runup elevations (which has no basis in physics whatsoever). So, I support publication subject to the points below being addressed by the authors.”

We thank the reviewer for providing complementary feedback on our research and for supporting publication following revisions. We agree that the submitted title is not entirely suitable, and have changed this to better represent the substance of the paper. The new title is:

“Predicting wave overtopping thresholds on coral reef island shorelines with future sea-level rise”

“Lines 18-21 imply use as a rapid assessment tool. As an engineer I think this needs some caveat. Real locations have complexity that is not included here, and thus relying on such a tool, or implying it is suitable for such an analysis in a local context, in my view risks overstating the protection afforded by reefs, particularly in extremes. Thus, I think emphasising the trajectories and general science outcomes is more beneficial.”

This is a fair point and we understand that our proposed method is limited in that it does not account for site specific nuances that have complexity beyond what we represent. To modify this sentence, we have changed the sentence to ‘provide a first-order’ assessment of vulnerability. This wording implies that we don’t account for all localised factors.

“The approach taken is well described and appropriate for the idealised scenario. But, it should be mentioned that the geometry considered is highly simplified as the authors know well, both for the reef and the "beach", where coral ridges and rubble are often built and maintained by waves as illustrated in the authors’ own publications, and there are no mangroves etc. There is no point glossing over this and it is not really provided in lines 422-430. What do the reef-beaches of the specific islands considered really look like? This does not negate the benefit of the study, but limits use to understanding broad scale trends, not site specific conditions, or only a very restricted set of sites.”

This is a good point. We agree that this simplification is not explicitly outlined. In addition to the changes we made to lines 125-132 in revised text (outlined in response to the editor comments) we have added the following new point to the list of limitations that addresses this comment:

“Using idealised island and reef morphologies omits the influence of micro-topography on wave processes on the reef flat and beach face. Accordingly, our

findings are based on a simplified geometry that represents the broad morphological characteristics of a reef.”

“In lines 70-78, it should be noted that the approach taken, of simulating a large range of idealised bathymetry and wave conditions using a physics-based wave model, was first proposed to study the protection afforded by coral reefs in this same context by Baldock et al. 2014; 2015 (Marine Pollution Bulletin), and included an estimate of how the morphological characteristics of the reef system might impact on sediment transport and beach erosion.”

We have added the two recommended papers, along with a few others that utilise a batch-processing approach to obtain synthesised knowledge from large sets of numerical modelling experiments. The following sentence has been added (revised text lines 73-75), with appropriate references: “Our approach builds on previous studies that utilised a wide range of idealised bathymetries and wave conditions to investigate the impacts of sea-level rise on reef coastlines”.

“The issue of how beach or beach crest morphology might change over time, and how this impacts the study conclusions, is briefly considered, lines 223-225, but again I believe the authors' own work on reef island morphology indicates that greater discussion of caveats is needed. These, or at least many other, islands have wave-built beach crests. There is little reason to suppose that they cannot keep up with rising sea levels (even if that means transgression). For example, during overwash, beach berms can easily keep up with the rise from neap to spring tides on natural beaches (Baldock et al, 20008, Geomorphology), so the response time is very rapid compared to sea level rise. Thus, it needs to be very clearly emphasised that the flood trajectories are for no change in current beach crest elevations. This issue has been discussed for reef crest elevation, but not sufficiently for berm crest elevation.”

As the reviewer points out, the submitted manuscript included a brief comment on the morphodynamic adaptation of reef shorelines to future sea level. We wanted to acknowledge this point but not discuss it in detail because understanding morphodynamic feedbacks is not the objective of this paper, but it is the specific focus of other work currently in preparation. To acknowledge this point we have made minor edits throughout the manuscript to ensure the

reader is aware that all overtopping trajectories and thresholds are clearly for an island system that undergoes no morphological change.

“The island inundation index is an interesting new function, but is inundation not a bit overstated? There is only a threshold, and the index does not indicate any magnitudes. Overtopping index?”

On reflection, we agree. We have changed the island inundation index to the reef island overtopping threshold (RIOT).

“Line 236. I think critical new insight is rather over doing it. Google can tell me the Marshall Islands are suffering from rising seas (or reef degradation?) and have been for some time. Islands that are 1-2m above sea level and currently often inundated at highest tides will clearly be impacted by 0.4m sea level rise. And to say that elevation is the single most critical value determining the magnitude of overtopping is rather obvious to a layperson, let alone coastal scientists.”

Contrary to what the reviewer suggests, we maintain that our manuscript generates critical new insight on the future vulnerability of reef coastlines to sea level rise. We have clarified the wording of the final paragraph to emphasise what we believe are critical new insights that result from our research. Importantly, this manuscript is the first to assess how interactions between primary boundary conditions (wave climate, sea level) and reef structural characteristics (depth, width and island height) influence overtopping on reef coastlines. While each of these variables is known to have a deterministic influence on wave overtopping, our research shows that it is possible to account for variability in all of these parameters.

To suggest that island vulnerability is controlled by elevation alone is misleading and significantly oversimplifies our outputs. One of our key outcomes is that future island vulnerability will vary spatially according to differences in morphology, which counters the assumption from previous work that case study insight is transferable. For example, our work shows that two islands of the same elevation may have significantly different vulnerabilities to future SLR because of differences in either reef depth, width or wave climate. Such

differences become significant in planning for an increase in wave driven flooding and elevated risk to communities.

“The reef geometry is an important control on what will happen, and the changes in nearshore conditions are a complex function of this geometry. This has been well illustrated in the previous work of Baldock et al. noted above, and they are not often monotonic functions. I think the authors could also illustrate how wave period changes the picture, since I would expect this to be very important, and it is very difficult to judge from figure 2 what that might be. Indeed, since neither parameter in figure 2 includes wave period, why is it there? A plot of how versus T for a given W/h would be more instructive.”

This is a good point and we acknowledge that the influence of wave period was not discussed in the manuscript but agree that it should be. The main reason for omitting wave period in Equation 1 was that our results show that wave period had a secondary influence on overtopping compared to other parameters. To amend this, we have included a new Supplementary figure (Sup Fig 2) as suggested, and added a new comment in the Methods section that outlines the influence of wave period when applying Equation 1. Supplementary Figure 2 shows that wave periods of 6 s are associated with significantly lower overwash levels compared to all other periods (as expected). There is some difference between moderate periods waves (8-12 s) and long period swell (14-16s) but as shown as a comparison in Supplementary Figure 2, this relationship between wave period and overwash depth is significantly weaker than the relationship between overwash depth and H_s or Z_r . Therefore, we chose to focus the analysis using only primary variables that are appropriate for most or all open ocean wave periods incident to reef islands.

Reviewer #3

"I recommend publications after addressing the following minor revisions:

Thank you for providing complementary feedback and a suggestions for improving our paper.

L8 Wave-driven flooding"

Typo corrected.

"L27 Consider rewording this sentence as too general (e.g. mangroves also regulate interactions between ocean waves and tropical coastlines)."

Sentence was changed to specifically address coral reef fringed shorelines.

"L38 Reword to include that SLR has already impacted low-lying coral reef islands (e.g. Albert et al. 2016. Interactions between sea-level rise and wave exposure on reef island dynamics in the Solomon Islands. Environmental Research Letters, 11(5), 054011)."

Sentence was modified to include the recommended reference:

"SLR is a serious climate change hazard that has already impacted some reef island systems¹⁴ and is predicted to impact all reef-fringed and atoll island shorelines in the coming decades^{15,16}, affecting an estimated global population of 197 million people⁴."

"L48 Suggest including: Barnard et al. (2015). Coastal vulnerability across the Pacific dominated by El Nino/Southern Oscillation. Nature Geosci, 8, 801-807. doi:10.1038/ngeo2539."

After carefully considering the recommended reference we decided it was not directly relevant for the suggested sentence or wider manuscript because it is not specific to reef environments.

"L194 What about scenarios of island or reef erosion?"

No change has been made.

Future scenarios of reef and island erosion are controversial but important topics that we believe are beyond the scope of our manuscript. Our focus was to present overtopping thresholds for the simplest scenario (no change) and to discuss how potential feedbacks that are relevant for management applications (accretion of the reef or island berm) may offset or delay future risk. In regards to reef accretion, our analysis was advised by well understood scenarios of reef responses to sea level rise (keep-up and catch-up modes). No such theory is available for representing scenarios of reef erosion and we believe that any future work on this topic needs to be driven by field observations and focused effort. Regarding changes in island elevation, most existing discussions on the morphological adaptation of islands and barriers to sea level rise show there is strong potential for accretion, with a lower probability of erosion (Baldock et al, 20008, *Geomorphology*). We chose to focus on the magnitude of island berm accretion required to prevent overtopping because this variable is a proxy for runup elevation and is relevant for guiding management options. We believe it is not possible to accurately represent scenarios of island erosion without undertaking additional research to quantify erosion rates which are not yet readily available. To address some of these existing research gaps, future work will focus on exploring reef island flooding in the context of sea-level rise and morphodynamic change.

"L372 Recent work by Harris et al. (2018). Coral reef structural complexity provides important coastal protection from waves under rising sea levels. *Science Advances*, 4(2). doi:10.1126/sciadv.aao4350) concludes that structural complexity on reef flats is more important than sea-level rise in terms of dissipation when it comes to average wave heights based on field data. Suggest rewording this sentence to reflect that not accounting for variability in roughness is a caveat of the model and needs to be incorporated in future studies."

As suggested by the reviewer we have added a comment in the methods to state that our omission of not representing variable friction magnitudes is a simplification that should be addressed in future studies: "The contribution of possible future reductions in friction associated with declining structural reef complexity to overtopping scenarios is a subject recommended for future study."